# The Perioperative Anesthetic Management of the Pediatric Patient with Special Needs: An Overview of Literature

**DOI:** 10.3390/children9101438

**Published:** 2022-09-21

**Authors:** Alessandra Ciccozzi, Barbara Pizzi, Alessandro Vittori, Alba Piroli, Gioele Marrocco, Federica Della Vecchia, Marco Cascella, Emiliano Petrucci, Franco Marinangeli

**Affiliations:** 1Department of Life, Health and Environmental Sciences, University of L’Aquila, 67100 L’Aquila, Italy; 2Department of Anesthesia and Intensive Care Unit, SS Filippo and Nicola Academic Hospital of Avezzano, 67051 L’Aquila, Italy; 3Department of Anesthesia and Critical Care, ARCO ROMA, Ospedale Pediatrico Bambino Gesù IRCCS, 00165 Rome, Italy; 4Department of Anesthesia and Critical Care, Istituto Nazionale Tumori—IRCCS, Fondazione Pascale, 80131 Naples, Italy; 5Department of Anesthesia and Intensive Care Unit, San Salvatore Academic Hospital of L’Aquila, 67100 L’Aquila, Italy

**Keywords:** perioperative medicine, anesthesia, children, psycho-physical disorders, pain, premedication, pediatrics, ketamine, dexmedetomidine, delirium

## Abstract

The perioperative management of pediatric patients with psycho-physical disorders with related relational and cognitive problems must be carefully planned, in order to make the entire hospitalization process as comfortable and as less traumatic as possible. This article reports an overview of the anesthetic management of non-cooperative patients between 6 and 14 years old. The pathologies most frequently responsible for psycho-physical disorders can be summarized into three groups: (1) collaboration difficulties (autism spectrum disorders, intellectual impairment, phobia); (2) motor dysfunction (cerebral palsy, epilepsy, other brain pathologies, neuromuscular disorders), and (3) craniofacial anomalies (Down syndrome, other genetic syndromes). Anesthesia can be performed safely and successfully due to careful management of all specific problems of these patients, such as a difficult preoperative evaluation (medical history, physical examination, blood sampling, evaluation of vital parameters and predictive indices of difficult airway) and the inapplicability of a “standard” perioperative path (timing and length of the hospitalization, anesthetic premedication, postoperative management). It is necessary to ensure a dedicated perioperative process that is safe, comfortable, tailored to specific needs, and as less traumatic as possible. At the same time, all necessary precautions must be taken to minimize possible complications.

## 1. Introduction

International scientific literature defines pediatric patients with “special needs” (SN) as children suffering from psycho-physical disorders with related relational and cognitive problems [1,2].

The perioperative management of these pediatric patients must be carefully planned, in order to make the entire hospitalization process as comfortable and as least traumatic as possible [3,4].

For pediatric patients with special needs, the standard perioperative path is inapplicable, mainly due to the surgical and anesthetic criticalities [5]. These patients have a much-increased perioperative risk due to their anatomical features, underlying disease and the coexistence of complex comorbidities [6]. The purpose of this article is to provide an overview of the literature in the anesthetic management of non-cooperative pediatric patients with SN, between 6 and 14 years old. The pathologies most frequently responsible for SN in children, were considered in this research.

## 2. The Non-Cooperative Pediatric Patient with Special Needs

Due to their clinical history, and the lack of collaboration, in pediatric non-cooperative patients with Special Needs (SN), the anesthesiological risk may result difficult to assess, despite the fact that they meet the criteria of the outpatient management applied for the patient without disabilities [7].

The peculiarities of the patient with SN are mainly represented by difficult preoperative evaluation (basic medical history, physical examination, blood sampling, evaluation of vital parameters and predictive indices of difficulty in managing the airways); the need to ensure a safe, comfortable and a least traumatic perioperative process as possible; and inapplicability of a standard perioperative path (timing of hospitalization, length of hospitalization, anesthetic premedication and postoperative management).

The international scientific literature recommends that the non-cooperative patient with SN must have easy access to the hospital, the possibility of performing blood chemistry and ECG tests, correct administration of the premedication in times and methods identified [8]. The preoperative interview must be performed by trained staff of the Preoperative Assessment Service. Anamnestic collection and physical examination should preferably take place in a dedicated room, where there are no disturbing elements and possibly without identification elements related to the healthcare environment.

### 2.1. The Pathologies Most Frequently Responsible for SN

These pathologies can be summarized in three groups:(1)craniofacial anomalies (Down syndrome, other genetic syndromes);(2)collaboration difficulties (autism spectrum disorders, intellectual impairment, phobia);(3)motor dysfunction (cerebral palsy, epilepsy, other brain pathologies, neuromuscular disorders).

#### 2.1.1. Down Syndrome (DS)

Down Syndrome (DS) represents 9% of all interventions in patients with SN and 1.25% of all anesthetic procedures [9]. Patients with DS have peculiar anatomical features such as a low set of ears, small teeth, flat nose (flat nasal bridge), stature delay, abnormal fingerprints and hypotonia [10]. They also present microbrachycephaly, short neck, macroglossia and protruding tongue, hypertrophic tonsils and adenoids, narrow subglottic area and prolapsed epiglottis [11]. Frequently, these patients are obese and obesity can add further anesthetic problems: omega epiglottis and tracheomalacia [12].

Craniofacial anomalies increase the incidence of intubation difficulties, perioperative airway obstruction and post-intubation spasms of the upper airways. Almost 50% of patients with DS have upper airway obstruction [11]. 

Patients with DS are at risk of atlantoaxial instability (AAI). This condition is also referred to as an atlantoaxial subluxation and it has been reported to occur in 6.8% to 27% of the DS population. The risks associated with AAI are neurological injury from excessive movement of the cervical vertebra impinging on and then damaging the spinal cord [11]. During the perioperative period, sedative and neuro-muscular blocking agents can promote uncontrolled movements between the first (atlas) and second (axial) cervical vertebra joint articulation, with consequent iatrogenic injuries. There is limited literature regarding the best algorithm, measurements, and criteria for screening children with DS for AAI. Some authors have recommended obtaining a full series of cervical spine images including flexion/extension lateral radiographs. Bouchard et al. underlined that a single neutral upright lateral radiograph was considered an efficient method for radiographic screening of cervical spine instability [13].

The intellectual disability of patients with DS can be mild or moderate, with a good understanding of language but with difficulty in verbal expression [14,15].

Comorbidities, sometimes highly disabling, are essentially represented by congenital heart disease (interventricular defect, the most common form; interatrial defect, found in 8% of cases; patency of the Botallo’s duct, in 7%; Fallot’s tetralogy in 1%) [16].

Table 1 summarizes the most frequent anesthetic problems of the patient with DS.

#### 2.1.2. Cri du Chat Syndrome

Cri du Chat Syndrome is a very rare congenital genetic syndrome with peculiar clinical characteristics such as microcephaly, enlarged nose root, epicanthus, hypertelorism, low-set ears, micrognathia and typical acute crying which gives the name of the syndrome [17,18]. The acute crying pitch depends on laryngeal anatomical abnormalities, including laryngomalacia with vocal cord paralysis, small and/or narrow larynx with diamond-shaped vocal cords, and epiglottis, which may be elongated, curved and fluctuating, or short and flaccid, hypoplastic and hypotonic [19]. Newborns can experience asphyxia, cyanotic crises, sucking difficulties and hypotonia. There is a severe delay in psychomotor development. Marked hypotonia at birth is observed, followed by hypertonia, psychomotor and speech retardation [20].

The main anesthetic problems are related to the difficulty of intubation due to poor visualization of the vocal cords, as well as the combination of several factors: laryngeal anomalies, micro-retrognathia, high palatine vault and limited cervical mobility, excessive salivation due to poor control of the swallowing (Table 2). If cardiac malformations are present, an important response to muscle relaxants responsible for hypotonia may occur, especially in infants [21,22,23].

#### 2.1.3. Autism Spectrum Disorders 

Autism Spectrum Disorders (ASD) cause impairment in social interaction, communication and isolation problems, behavioral disturbances, disorders of the sensory sphere, as well as significantly altered motor skills [24]. The problems of communication and social interaction are linked to a limited vocabulary, language deficits (deficit of acquisition, echolalia, occasional mutism), the inability to understand figurative language and the meaning of conversations [25]. Some patients exhibit behavioral disturbances, such as self-mutilation, aggression, psychotic disturbances, resistance to change, gesture repetition, mechanical memory, excessive emotional reactions, and motor awkwardness, stereotyped patterns of movement, language or manipulation of objects, excessive adherence to a daily routine made up of motor or verbal rituals [26]. Sensory disturbances are represented by inadequate responses to tactile and proprioceptive stimuli and by an altered perception of pain [27]. The painful experience can cause communication and social interaction difficulties [28]. The autistic patient may have a global delay in psychomotor development associated with intellectual disability, sleep, and eating disorders. Epilepsy or electroencephalographic abnormalities, psychiatric disorders, and behavioral alterations, such as hyperactivity and impulsivity, anxiety, obsessive-compulsive disorders, bipolar disorder, and depression are also very common in ASD children [29,30].

Table 3 summarizes the most frequent anesthetic problems of the patient with ASD.

The precautions to be adopted for the correct management of the patient with ASD, are shown in Table 4.

#### 2.1.4. Infantile Cerebral Palsy

Infantile cerebral palsy (CP) includes several clinical manifestations, from monoplegia with normal cognitive function to spastic quadriplegia with mental retardation [31]. The causes are multiple and are responsible for damage to the central nervous system that can occur in the prenatal, perinatal or postnatal period [32]. The Spastic Form, associated with intellectual disability and epilepsy, presents an altered control of the buccal muscles, tongue and pharynx resulting in various degrees of difficulty in feeding (nasogastric feeding or gastrostomy are often necessary). The dyskinetic form can be associated with dystonia, athetosis and chorea; epilepsy often concomitates, while the IQ is usually normal. Balance disorders, cerebellar tremors, language difficulties, epilepsy and cognitive deficits are associated with the ataxic form. The Mixed Form includes the remainder of the cases where more subtypes are associated with the related clinical manifestations [33]. There are clinical and anatomical features that we should consider before surgery in general anesthesia, such as: the presence of gastroesophageal reflux which, together with problems of esophageal motility and anomalies of the lower esophageal sphincter, increase the risk of aspiration gastric contents; sialorrhea and poor swallowing control which can lead to difficulties in mask ventilation and visualization of the glottis during oro-tracheal intubation [34]. Recurrent lung infections in conjunction with chest deformities (scoliosis) can cause chronic cardiopulmonary impairment. Table 5 summarizes the anesthetic problems of the patient with infantile cerebral palsy.

#### 2.1.5. Epilepsy

Epileptic patients usually take long-term anti-epileptic therapy; monotherapy antiepileptic drugs are often effective in seizure control and a significant number of these patients require polypharmacological anti-comital therapy [35]. It can be associated with the treatment of concomitant pathologies. Epileptic patients are exposed to an increased risk of clinically significant drug interactions, especially with traditional anti-epileptic drugs [36]. With regards to the interactions examined between new generation anti-epileptic drugs and drugs used for the treatment of related non-epilepsy disorders, pharmacokinetic interactions have always referred to hepatic enzyme induction or inhibition [37]. Furthermore, pharmacological dosages of anesthetic drugs, such as hypno-inducers was highlighted. The required dose of Propofol is significantly lower and the recovery time was longer [38].

Table 6 summarizes the indications to be followed in the anesthetic management in the patient with epilepsy.

## 3. Perioperative Anesthetic Phases

Table 7 summarizes the anesthetic phases in the management of the pediatric patient with Special Needs (SN).

### 3.1. Preoperative Anesthetic Visit 

During this stage, it is essential to identify the caregiver figure (CG) or the leading parent (LP). The first contact with the CG or LP can be by phone or at the pre-operative assessment service.

The non-cooperative patient with SN (difficult access to hospital, difficulty or inability to perform electrocardiogram and/or blood tests, inability to perform physical examination and evaluation of predictive indices of difficulty in managing airways) must be examined, after parents’ agreement, with the mediation of the surgeon [39,40].

In the absence of important comorbidities of the anesthetic interest, preoperative examinations can be performed after the induction of general anesthesia. If possible, in case of comorbidities, a recent anamnestic documentation, not exceeding two years, is required, including the documentation relating to routine blood chemistry tests, the list and dosage of drugs in use (e.g., lithium, anticonvulsants, antipsychotics), instrumental examinations (chest X-ray, brain MRI, EEG) in possession of the parents/guardians [41]. 

It is important to underline that prolonged administration of antiepileptic drugs is associated with several drug interactions, due to multiple factors operating alone or in combination. Specifically for anesthesia and critical care, sensitivity and resistance to non-depolarizing neuromuscular blockers (NDNMBs) after acute and long-term administration of antiepileptic drugs, have been well described. This can result from hepatic drug metabolism induction, increased protein binding of the NDNMBs and/or upregulation of acetylcholine receptors. A neuromuscular blocking enhancement due to pre- and post-junctional direct effects can occur after an acute administration of antiepileptic drugs [42].

Patient documentation must be requested at the time of the first telephone contact with the CG or LP. It is often difficult to access the hospital, perform the electrocardiogram, blood tests and a physical examination, the evaluation of predictive indices of difficulty in ventilation/intubation which facilitate a safe and provident management of the airways. The anesthetist, therefore, is at a potential risk of difficult intubation and/or ventilation due to anatomical maxillofacial and upper airway malformations which are often present. The preoperative evaluation must be limited to the collection of the pathological anamnesis with the help of the LP or the CG. The pathological history allows for the identification of comorbidities (cardiological, respiratory, metabolic, neurological) and anesthetic risk factors (home therapy, airway management, myopathies, epilepsy, severe cervical and dorsal-lumbar spine deformities); the vision of any previously requested documentation and the home therapy (drug interactions with anesthesia drugs) [43]. A form with an accurate drug history must be included in the anesthetic record chart. It allows for the avoidance of drug interactions that could interfere with all stages of anesthesia and in particular, with the patient’s awakening. Difficulty in finding a venous access should be noted. It is essential that the surgeon and the anesthesiologist have already discussed each individual case at this stage in order to plan the surgery and prepare special measures. The LP or the CG may also, at this stage, indicate habits, special needs and any alternative communication channels used by the patient (specifically relating to the communication of discomfort, agitation, fear, pain or a state of well-being mostly useful in the awakening phase). The LP or CG play an important role in mediating relationships with the patient and they help to build an empathic relationship between patient and anesthetist. Familiarization with the healthcare environment is another important element, however, it requires more than just one meeting and this is not always possible. If possible, it is advisable to fill in, with the help of the LP or CG, an additional preoperative evaluation form that helps to understand the particular needs and habits of each patient.

### 3.2. Establishment of a Dedicated Surgical Session and Hospitalization

It is essential to establish a fixed day for the operating session in order to organize the procedure [9]. In the same way, any change in the day dedicated to the operating session must be communicated at least one month before. The nursing, auxiliary and medical staff must be fully dedicated to the activity carried out in the ward and in the operating room and they should be adequately trained [44]. For each session, an operating list that respects the time assigned to that session must be prepared, compiled by evaluating the surgical and anesthetic times required for each procedure and based on the complexity of the patient [45].

### 3.3. Preparation for Admission

All patients should be hospitalized on the same day of the surgery, in a reserved room in the surgical department, in order to facilitate treatment and assistance by the medical and the nursing staff dedicated to them. Admissions should be according to the order of the list and to the times provided for each procedure to avoid excessive hospitalization time, which leads to lack of care and stress for the patient [46].

### 3.4. The Anesthetic Premedication

Hospitalization and surgery can cause significant stress and anxiety in children. Induction of anesthesia may be the most exhausting procedure that a child experiences during the entire perioperative period [47]. Premedication is recommended. The premedication should be chosen on the basis of personal preferences because there is no sufficient literature to be able to choose based on evidence. Adequate preoperative sedation allows these patients and their families a more serene hospitalization. The main purpose of premedication in the pediatric patient, especially in the patient with SN, is anxiolysis which can facilitate separation from the LP or CG and make the induction of anesthesia easier. In addition, adequate premedication includes: amnesia, the prevention of physiological stress, the reduction in the demand for anesthetics, the antiemetic effect, vagolysis, reduction in secretions and analgesia. It is advisable during the anesthetic visit to establish with the LP or the CG the methods of administration of the pre-anesthetic drug: the type, the dosage, the route of administration, the time of administration, and the titration of the drug, in order to achieve a degree of sedation that allows for venous cannulation and intravenous or inhalation induction. In these patients, the induction of intravenous anesthesia is not always possible. It’s often administered the “steal inhalational induction”. This induction involves the arrival of the patient in the operating room moderately sedated thanks to anesthetic premedication [48]. The drugs currently used are listed in Table 8. The following are considered first choice drugs: midazolam, clonidine and dexmedetomidine; ketamine is a second-choice drug.

All drugs used may produce sedation and respiratory depression and should always be administered under supervision and monitoring. Devices for supplemental oxygen administration and support for ventilation and resuscitation should be readily available. Inhalation of the nebulized drug is an alternative method of administration that is relatively easy to set up, does not require venous access and is associated with a high bioavailability of the drug employed [51]. Abdel Ghaffar observed that children premedicated with nebulized inhaled dexmedetomidine (2 μg/kg) have more satisfactory sedation scores, greater mask acceptance and shorter recovery times than who received nebulized ketamine (2 mg/kg) [52] or midazolam (0.2 mg/kg) [53]. Premedication with dexmedetomidine reduces the incidence of postoperative psychomotor agitation.

#### 3.4.1. The Routes and the Times of Administration of the Premedication

The routes of administration are the oral route in first choice and the intramuscular route in 2nd choice (after applying the patch of lidocaine and prilocaine, EMLA^®^). EMLA^®^ provides adequate topical anesthesia also for the insertion of an intravenous catheter and must be applied one hour before venous cannulation. In non-cooperative patients, premedication, by the oral or intramuscular route, must be administered by the ward nurses, as indicated on the anesthetic chart, 45 min before being transferred to the operating room. In non-collaborating patients, if possible, premedication should be administered 1 h before surgery by the oral route. It is advisable to combine sugar with the oral sedative or dilute the drug with a patient’s favorite beverage in a teaspoon (clear liquids) [54]. After 45 min, the patient must be led by the ward’s dedicated nursing staff, also accompanied by the LP or CG, into the antechamber of the Operating Room. For this purpose, on the days of the operating session, a dedicated ward nurse exclusively takes care of patients with SN, with: the preparation, the correct premedication administration, the transport to the operating room, the delivery of the patient to the dedicated room staff and the return to the department at the end of the session. It is necessary that all the personnel involved are informed about their duties and those of other professional figures.

#### 3.4.2. Sedation Assessment Score

In the department, after the first administration of a sedative drug, the depth of sedation must be measured using the following score: 0 = patient completely awake; 1 = patient calm, awake, but not transportable from the antechamber to the operating room; 2 = sleepy and calm patient, easy to awake, collaborative in preoperative anesthetic procedures; 3 = sleepy patient, difficult to awake, eupneic, normotensive. This evaluation must also be repeated in the antechamber of the operating room.

#### 3.4.3. Premedication Drugs

##### *Midazolam,* *Temazepam,* *and* *Lorazepam*

Midazolam is a water-soluble benzodiazepine and the most commonly used sedative in premedication in children. Benefits include rapid, reliable onset and antegrade amnesia with minimal respiratory depression. It is generally administered orally at a dose of 0.3–0.75 mg/kg, up to a maximum of 20 mg, sedation and anxiolysis are reliably achieved within 20 min. In addition to the oral route, it can alternatively be administered intranasally (0.3 mg/kg), rectally (0.5 mg/kg) or sublingually (0.3 mg/kg). Postoperative sedation is a side effect, especially after short surgeries. Oral midazolam may not produce sedation in 20% of patients. In older children, Lorazepam and Temazepam are useful anxiolytics. The oral tablet of Lorazepam at a dose of 0.025–0.05 mg/kg administered 60 min previously has a duration of action of 12 h. Temazepam was used orally at a dose of 0.3–0.5 mg/kg 1–2 h before the induction. Diazepam is an unpopular choice for premedication in children; its metabolite Desmethyldiazepam has similar pharmacological activity to the parent compound. Immature liver function prolongs the half-life.

##### *Triazolam* 

Triazolam is a short-acting benzodiazepine that possesses dose-dependent anxiolytic, sedative, amnestic, anticonvulsant, muscle relaxant, and hypnotic properties [55]. During the surgical removal of third molars, 0.25 mg of orally administered triazolam 45 to 60 min before intravenous sedation can provide anxiety relief while maintaining protective reflexes and responsiveness. This type of sedation assists SN patients’ ability to tolerate the stress associated with dental treatment [55].

Triazolam at 0.375 mg has a strong and significant amnesic potential. This effect could be helpful for SN patients with anxiety related to the operating room before surgery [56]. Triazolam at 0.5 mg has been demonstrated to have a stronger sedative and anxiolysis effect compared to 0.25 mg but induces prolonged cognitive deficits with hemodynamic changes [56]. Indeed, triazolam can be used cautiously at specific doses or in combination with other sedative drugs in SN children with hepatic dysfunction. Hepatic dysfunction can lead to prolonged sedation or impairment of psychomotor function caused by high plasma concentrations and high sensitivity to triazolam [57].

##### *Clonidine* *and* *Dexmedetomidine*

Alpha 2-adrenergic agonists are used before surgery to reduce anxiety in non-cooperative children [58]. This group of drugs also provides clinically relevant benefits in reducing the need for rescue analgesia, reducing agitation on awakening, postoperative nausea and vomiting (PONV), and post-operative shivering [59]. Clonidine can be administered orally (3–4 μg/kg) or intranasally (2 μg/kg). Nasal Clonidine is not associated with nasal burning.

Premedication with Clonidine is superior to Midazolam in terms of sedation production, decreased postoperative pain, PONV and agitation [60].

Clonidine has a long onset time (45 min) but offers advantages such as prolonged analgesia and intraoperative saving of anesthetics, especially in surgery with significant postoperative pain.

Dexmedetomidine has a shorter half-life (approximately 2 h in children) than Clonidine [61,62].

Compared to Midazolam, Dexmedetomidine produces satisfactory sedation after separation from parents and greater acceptance of the face mask.

Oral administration is associated with poor bioavailability. Intranasal Dexmedetomidine was used satisfactorily at a dose of 1 μg/kg administered 45–60 min before the induction. Limitations for its use include long onset times (30 min) and bradycardia and hypotension with higher doses [61,63].

##### *Ketamine* 

It has long been used in premedication. It can be administered orally (5–8 mg/kg), intramuscularly (4–6 mg/kg) or iv (1–2 mg/kg) [64]. Benefits include its analgesic properties and the ability to cause sedation not associated with respiratory depression [65]. The undesirable effects are represented by: increased salivation, delirium upon awakening and prolongation of cognitive recovery. Due to the availability of new agents with fewer side effects, the role of Ketamine is generally reserved for the older child, with developmental delay or with Autism Spectrum Disorders (ASD). In these patients a sedative dose of intramuscular Ketamine can be effectively administered (deltoid injection works within 2–3 min) [66].

##### *Melatonin* 

Melatonin is reported to have a protective effect against delirium with sedative effects and decreasing sedative consumption. It is similar to benzodiazepines in reducing preoperative and postoperative anxiety in adults [67]. Melatonin can be used as a sedative agent in surgical patients with SN regardless of the type of surgery [68] (A dose of 0.5 mg/kg orally administered 30 min before surgery has been shown to reduce agitation in non-cooperative patients. However, this drug cannot reduce agitation and delirium during in hospital stay, especially during perioperative period. It is important to underline that the optimum dosage and formulation of melatonin, and treatment duration still remain uncleared and open.

##### *Electric/Virtual* *Sedation* *and* *Non-Pharmacological* *Interventions*

Virtual reality and other electronic distraction techniques may have multiple clinical applications in child sedation. “Non-aversive” techniques can serve to relieve harmful behaviors in order to safely provide quality care, minimize or eliminate the need for aversive measures, and help in cooperation development [69].

Non-pharmacological interventions such as parental acupuncture, clowns/clown doctors, playing videos of the child’s choice, low sensory stimulation, and hand-held video games have promising potential for reducing anxiety and improving cooperation during preoperative sedation [70]. However, these methods do not usually provide adequate relief and thus, pharmacological sedation and analgesia are necessary [71]. Wang et al. suggested that clinical programs that supplement virtual reality can be considered in pediatric periprocedural care [71]. By definition, pediatric patients with “special needs” (SN) are children suffering from psycho-physical disorders and relational and cognitive problems [1,2]. In addition, due to non-cooperative aspects along with immature cognitive skills, stress coping inabilities, and low or negligible attention spans, such patients are likely to have a poor adaptive response following anxiety [69]. As a result of these reasons in SN children, differently from non-cooperative children without SN, non-pharmacological management strategies may often prove inadequate or inappropriate to overcome such behaviors during the anesthesia perioperative phase. The wide use of smart phone applications can easily appeal to various characteristics of children and have been shown effective on relieving anxiety in children and their parents during the waiting period before operating room entry [72]. A strong correlation was found between smartphone application use on reducing perioperative anxiety and sedation on characteristics of psychosocial and psychophysical development. Some smartphone applications can stimulate child curiosity and in turn lower anxiety. However, younger or SN children are more vulnerable to anxiety due to their immature development and smartphone applications can paradoxically increase the perioperative agitation and anxiety [72]. Therefore, even though electric/virtual sedation seems to have a promising potential for reducing anxiety and improving cooperation in children with SN during the preoperative sedation, there remains a need for solid scientific literature.

##### *Intraoperative* *Management*

The non-cooperative and transportable patient (sedation score equal to 2) is accompanied to the operating room, always in the presence of the CG and LP, entrusted to the dedicated nursing staff. There is no specific contraindication to the use of anesthetic agents, such as Propofol, Sevoflurane and Desflurane. In uncooperative patients or in patients with physical disabilities, venous access for the induction of anesthesia is the main challenge and cannot always be ensured while the patient is still awake. In these cases, Sevoflurane can be used to induce safely and effectively anesthesia [28,33,38].

In the operating room, the monitoring of vital parameters, the depth of anesthesia using the Biospectral Index [73], and the myoresolution using TOF Watch are performed. After that we proceed with the induction of general anesthesia by the intravenous route, if it was possible to obtain venous access, or by inhalation, through inhalational steal induction [74]. In the latter case, when the patient is unconscious (BIS 40–60), drugs are administered intravenously (muscle relaxant and central analgesics, gastro-protectors and cortisone, if required). It is advisable to administer an analgesic before awakening, the choice of the drug will be based on the type of surgery.

The interaction between home drugs taken by the patient and anesthetics affects the intraoperative period. The dose of Propofol required for patients using antiepileptic drugs is higher than the dose for healthy patients [38]. Monoamine oxidase inhibitors (MAOIs) or selective serotonin reuptake inhibitors (SSRIs) have an increased risk of hypo/hypertension and prolonged awakening after general anesthesia [75]. Risperidone can cause hypotension under general anesthesia and can lead arrhythmias. It may be appropriate to discontinue long-acting antipsychotics and switch to short-acting or lower dose antipsychotics after consultation with the psychiatrist. During anesthesia, Clozapine can cause agranulocytosis, hyperthermia, cardiac conduction problems and hypotension. Psychostimulants can: increase the dose of sedative needed during anesthesia and the risk of hypertension and arrhythmias, lower the seizure threshold and interact with vasopressors [76].

Congenital heart disease affects approximately 50% of patients with DS and this can increase the risk of complications during anesthesia, such as bradycardia. The incidence of bradycardia in DS patients is approximately 3.7%, significantly higher rate than the approximately 0.36% found in healthy patients [77]. Congenital heart disease can also lead to other complications such as pulmonary hypertension [78,79,80].

Airway management can be difficult for patients with cerebral palsy due to excessive secretions and the risk of aspiration during anesthesia due to gastroesophageal reflux [81,82].

These patients also have a greater risk of hypoxia during general anesthesia [75]. In addition, about 30% of patients with cerebral palsy take home anti-comital therapy with an increased risk of drug interactions.

Table 9 summarizes the most frequent intraoperative challenges for an anesthetist in cerebral palsy patients.

### 3.5. Awakening and Postoperative Management

Anti-epileptic drugs cause a prolonged awakening time from anesthesia, but these drugs should not be discontinued during the preoperative in order to reduce perioperative seizures [39,82,83,84].

In Higuchi’s study, the awakening time in patients with intellectual disabilities is significantly longer than in patients without intellectual disabilities. Furthermore, the BIS Spectral Index is lower and the dose of Propofol and Remifentanil is significantly lower [85].

Prolonged awakening time from anesthesia appears to be correlated with frequent cholinergic dysfunction in patients with mental disabilities [85,86].

Therefore, it is important to meticulously monitor the patient during awakening because prolonged awakening time is associated with greater difficulty in maintaining patent airways due to the presence of craniofacial abnormalities and drug interactions.

#### Postoperative Complications

The risk of postoperative anesthetic complications in patients with SN depends on the patient’s ASA classification, clinical condition, type of anesthetic used and type of surgical procedure performed [87]. About 4.2% of cases have moderate complications, such as hypotension. Airway obstruction is the most common complication, followed by nausea and vomiting [81,87]. Yumura and Coll. reported an incidence of postoperative nausea and vomiting in patients with intellectual disabilities of 5.6%, a percentage higher than the general population [87,88]. Lim and Coll. found that 44.4% of patients with cerebral palsy have complications secondary to difficult airway management. In particular, there was an incidence of respiratory problems of 30.4% in patients with ASD, 29.2% in patients with DS and 17.1% in patients with intellectual disabilities.

## 4. Conclusions

General anesthesia is the most suitable type of anesthesia in pediatric patients with Special Needs (SN), although anesthetic complications are more frequent in these patients, mainly due to comorbidities, taken drugs and anatomical peculiarities. Anesthesia can be performed safely and successfully thanks to a careful management of all phases, which takes care of the specific problems of these patients, such as a difficult preoperative evaluation (medical history, physical examination, blood sampling, evaluation of vital parameters and predictive indices of difficult airway) and the inapplicability of a “standard” perioperative path (timing and length of the hospitalization, anesthetic premedication, postoperative management). It is necessary to ensure a dedicated perioperative process that is safe, comfortable, tailored to specific needs, and as less traumatic as possible. At the same time, all necessary precautions must be taken to minimize possible complications. This article reports an analysis, through the evaluation of the literature, of the main evidence useful for the correct anesthetic management of pediatric patients with Special Needs (SN).

## Figures and Tables

**Table 1 children-09-01438-t001:** Recurrent anesthetic problems in the patient with DS.

a. Obesity
b. Difficulty of venous access
c. Craniofacial anomalies (tonsillar hypertrophy, small subepiglottic area, omega or prolapse of epiglottis, macroglossia, atlanto-occipital instability)
d. Congenital heart anomalies
e. Epilepsy
f. Anesthesiological complications (bradycardia, airway obstruction, post-intubation croup)

**Table 2 children-09-01438-t002:** Recurrent anesthetic problems of the patient with Cri du chat syndrome.

Difficulty in intubation poor visualization of voice chords (laryngeal anomalies, micro-retrognancy, high palatine vault and limited cervical mobility)
Excessive salivation due to poor control of swallowing inhalation
Cardiac malformations
Accentuated response to myorelaxants for hypotonia (especially in infants and in the presence of associated heart malformations)

**Table 3 children-09-01438-t003:** Recurrent anesthetic problems in the patient with ADS.

Involvement of the leading parent or of the care giver
Need for flexibility in the choice of anesthetic drugs
Pharmacological interactions between anesthetics and home therapy
Comorbidity (spectrum patients)
Altered tactile sensitivity (in the pediatric patient)

**Table 4 children-09-01438-t004:** Precautions to be adopted for the correct management of the patient with ASD.

DO’S	DON’TS
Recognize the signs that indicate an increasing level of stress.	Start stress triggering factors (force maneuvering, lack of communication, focus on speed).
Use stress-reducing activities, along with the pl or gc (use of special interests as a distraction).	Unrespect the patient’s times to adapt
Use a quiet, assertive voice with a focus on distraction (special interest).	Use anger to communicate, raise your voice.
Provide low sensory stimuli, always use tranquility and firmness.	Giving into emotion, could be counterproductive.

**Table 5 children-09-01438-t005:** Recurrent anesthetic problems in the patient with Cerebral Palsy.

Difficulty in swallowing (airway inhalation)
Gastroesophageal reflux
Small for age patient
Difficult positioning (nervous and/or muscle damage)
Resistance to non-depolarizing curare
Close correlation between CP severity and incidence of postoperative complications (hypothermia, hypotension)
ASA physical status 2 o > 2: (epilepsy, hypotony of the upper airways)
Possibility of sedation (based on physical conditions, type of procedure, duration of the surgical operation)

**Table 6 children-09-01438-t006:** Anesthetic indications in the patient with Epilepsy.

Identification of the type, frequency, severity, seizures and triggering factors of Epilepsy
Maintenance of therapy in the perioperative period
Knowledge of the pro or anti-convulsant properties of anesthesia drugs
Crisis prevention, intra and postoperative
Reduction in anesthesia drugs due to the sedative effect of anti-comital therapy or inhibition of metabolic enzymes
Acid-base balance control (ketoacidosis from a high-fat ketonic diet)

**Table 7 children-09-01438-t007:** Anesthetic phases.

1	2	3
Preoperative anesthetic visit	Establishment of a dedicated operating session	Preparation for admission
4	5	6
Premedication	Establishment of a dedicated operating session	Postoperative management and awakening with any post-op complications.

**Table 8 children-09-01438-t008:** Commonly used medications for premedication in children.

DRUG	ROUTE OF ADMINISTRATION	DOSE	TIME TO EFFECT
Benzodiazepines	IN	0.3 mg/kg	
Midazolam	IV	0.05–0.1 mg/kg	10 min
Lorazepam	PR	0.5 mg/kg	2–3 min
Temazepam	PO	0.025–0.05 mg/kg (max 4 mg)	30 min
Triazolam	PO	0.25–0.5 mg	50 min
Alpha-agonists	IN	2–4 µg/kg	30–60 min
Clonidine	PR	2.5–5 µg/kg	
Dexmedetomidine	IN	1–2 µg/kg	
NMDA Antagonist	IM	4–6 mg/kg	3–5 min
Ketamine (*)	IV	0.5–1 mg/kg	1 min
Chloral Hydrate	PO	25–75 mg/kg (max 2 g)	30–45 min
	PR		
Melatonin	PO	0.5 mg/kg	20–30 min

(* [49,50]) Administration of drugs. IN: intranasally; IV: intravenously; PR: rectally; PO: orally.

**Table 9 children-09-01438-t009:** Intraoperative challenges for an anesthetist in cerebral palsy patients.

VENIPUCTURE (STEAL INDUCTION)
Monitoring during anesthesia
Drug interactions
Drug Choice for anesthesia
Airway management (excessive secretions, inhalation, intubation) difficulty.
Awakening
Length and kind of the procedure
Post-operative management (pain, management, post-operative nausea and vomiting, shivering)

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
