# Peer review of "The Perioperative Anesthetic Management of the Pediatric Patient with Special Needs: An Overview of Literature"

_children, 2022, doi:10.3390/children9101438_

Round 1

Reviewer 1 Report

This is a review article on the special-needs or developmentally delayed pediatric patient.  Given its length, a significant amount of time has been spent on this manuscript.  However, this paper has multiple weaknesses including

1) Requires major English editing.  There are a significant amount of run-on sentences and grammatical errors throughout the paper.  To begin, I recommend using the term "non-cooperative" instead of "non-collaborative" throughout the paper

2)  Requires major formatting changes.  The tables are not well made, are a mix of all caps and lower case with no clear explanation why, and often do not line up in appropriate rows.  Overall makes the manuscript not well edited nor prepared

3) Topic is not "state of the art" and very little recent literature has  been added on this topic.  Unclear about what this paper adds.

4) Seems disorganized with an overview of some syndromes then a summary about stages of anesthesia/procedures that then circles back to some specifics about the syndromes in later portions.  There are summaries that I don't believe are necessary - basic descriptions of medications should not be included in an anesthesia paper as mechanisms should already be known.  I found the paper difficult to follow.

5) Paper does not mention electric/virtual sedation which is the one recent development in the field over recent years.  This shows the authors' overall lack of knowledge and preparation in building this manuscript.  Papers regarding the use of virtual reality, iPads, and other electronic distraction techniques would be an essential part of any review on this topic

Author Response

Comments and Suggestions for Author

Reviewer 1

This is a review article on the special-needs or developmentally delayed pediatric patient. Given its length, a significant amount of time has been spent on this manuscript. However, this paper has multiple weaknesses including

1) Requires major English editing. There are a significant amount of run-on sentences and grammatical errors throughout the paper. To begin, I recommend using the term "non-cooperative" instead of "non-collaborative" throughout the paper

We appreciated your suggestions and constructive criticism. The manuscript has been reviewed for grammatical errors.

2) Requires major formatting changes. The tables are not well made, are a mix of all caps and lower case with no clear explanation why, and often do not line up in appropriate rows. Overall makes the manuscript not well edited nor prepared

We apologize. The tables have been modified in order to make the manuscript well-edited and prepared.

3) Topic is not "state of the art" and very little recent literature has been added on this topic. Unclear about what this paper adds.

Thank you for these suggestions. The term “state of art” has been deleted. This article reports an overview of the anesthetic management of non-cooperative pediatric patients, between 6 and 14 years old. In this research, we have focused on the pathologies most frequently responsible for special needs in children. Further explanations are located at the end of the introduction section.

4) Seems disorganized with an overview of some syndromes then a summary about stages of anesthesia/procedures that then circles back to some specifics about the syndromes in later portions. There are summaries that I don't believe are necessary - basic descriptions of medications should not be included in an anesthesia paper as mechanisms should already be known. I found the paper difficult to follow.

Thank you for arising this concern. This paper analyzed the most common pathologies responsible for special needs in children, as well as the stages of anesthesia/procedures.

We then recalled readers' attention to some specific points about the syndromes and deleted some basic descriptions of medications, as you suggested.

We are confident that the paper is now easier to follow.

5) Paper does not mention electric/virtual sedation which is the one recent development in the field over recent years. This shows the authors' overall lack of knowledge and preparation in building this manuscript. Papers regarding the use of virtual reality, iPads, and other electronic distraction techniques would be an essential part of any review on this topic.

We apologize if our manuscript did not mention electric/virtual sedation. We are perfectly aware that virtual reality and other electronic distraction techniques may have multiple clinical applications in children’s sedation. "Non-aversive" techniques can frequently serve to obtund interfering and potentially harmful behaviors to safely permit quality care, minimize or eliminate the need for aversive measures, and help in cooperation development (Nathan JE. Effective and safe pediatric oral conscious sedation: philosophy and practical considerations. Alpha Omegan. 2006;99(2):78-82. doi: 10.1016/j.aodf.2006.06.010).

Non-pharmacological interventions such as parental acupuncture, clowns/clown doctors, playing videos of the child's choice, low sensory stimulation, and hand-held video games have promising potential for reducing anxiety and improving cooperation during preoperative sedation (Manyande A, Cyna AM, Yip P, Chooi C, Middleton P. Non-pharmacological interventions for assisting the induction of anaersthesia in children. Cochrane Database Syst Rev. 2015 Jul 14;2015(7):CD006447. doi: 10.1002/14651858.CD006447.pub3), but these methods are often inadequate for treating pain and anxiety, so pharmacological sedation and analgesia are necessary, as noted by Jia MQ et al. (Jia MQ, Yuan XG. Current status and research advantages on drug sedation and analgesia inburn children. Zhonghua Shao Shang Za Zhi. 2022 Feb 20;38(2):190-195. doi: 10.3760/cma.j.cn501120-20200908-00404). Clinical programs that reliably provide access to virtual reality, can be considered such as a supplement in pediatric periprocedural care, as Wang et al suggested (Wang E, Thomas JJ, Rodriguez ST, Kennedy KM, Caruso TJ. Virtual reality for pediatric periprocedural care Curr Opin Anaesthesiol. 2021 Jun 1;34(3):284-291. doi: 10.1097/ACO.0000000000000983). By definition, pediatric patients with "special needs" (SN) are children suffering from psycho-physical disorders with related relational and cognitive problems (Caicedo, C. Families with Special Needs Children: Family Health, Functioning, and Care Burden. J Am Psychiatr Nurses Assoc 2014, 20, 398–407, doi:10.1177/1078390314561326; Huang, L.; Freed, G.L.; Dalziel, K. Children With Special Health Care Needs: How Special Are Their Health Care Needs? Acad Pediatr 2020, 20, 1109–1115, doi:10.1016/j.acap.2020.01.007). In addition, because of lacking in cooperative ability, immature cognitive skills, a highly restricted range of coping abilities, a brief or negligible attention spans, and virtually no experience coping with stress, they can be especially prone to maladaptive responses to anxiety-provoking situations (Nathan JE. Effective and safe pediatric oral conscious sedation: philosophy and practical considerations. Alpha Omegan. 2006;99(2):78-82. doi: 10.1016/j.aodf.2006.06.010). Because of these reasons in SN children, differently from non-cooperative children without SN, non-pharmacological management strategies may often prove inadequate or inappropriate to overcome resistive or uncooperative behaviors during the anesthesia perioperative phase. The behavioral intervention program using a smartphone application effectively relieved anxiety in children and their parents, during the waiting period before operating room entry, because it can easily appeal to various characteristics of children, and various smartphone applications are easily obtainable (Lee JH, Jung HK, Lee GG, Kim HY, Park SG, Woo SC. Effect of behavioral intervention using smartphone application for preoperative anxiety in pediatric patients. Korean J Anesthesiol. 2013 Dec;65(6):508-18. doi: 10.4097/kjae.2013.65.6.508). A strong correlation was demonstrated between smartphone applications' usefulness in perioperative anxiety reduction and sedation providing, and characteristics of psychosocial and psycophysic developmental stage. Some smartphone applications can stimulate children’s curiosity, resulting in a reduction of anxiety. However, younger or SN children are more vulnerable to anxiety due to their immature development. Consequently, smartphone applications can paradoxically increase the perioperative agitation and anxiety, then resulting complex to control (Lee JH, Jung HK, Lee GG, Kim HY, Park SG, Woo SC. Effect of behavioral intervention using smartphone application for preoperative anxiety in pediatric patients. Korean J Anesthesiol. 2013 Dec;65(6):508-18. doi: 10.4097/kjae.2013.65.6.508).

The electric/virtual sedation seems to have a promising potential for reducing anxiety and improving cooperation during preoperative sedation, but scientific literature lacks of a clear consensus about its usefulness in children with SN.

Therefore, we decided to neglect adding this paragraph in the original version of the manuscript. However, in the revised version of the manuscript, you can find this paragraph.

We have added this paragraph according to your indications.

Reviewer 2

Thank you for appreciating our work. We are grateful for your constructive criticism and suggestions. We are confident that your suggestions have improved our manuscript.

Page 2, line 50- should be “as least traumatic as possible” rather than “less” (same page 2, line 64)

The term “less” has been replaced with “least”, as you suggested. Please see page 2, lines 53 and 72.

  • page 2, line 57- is this implying that SN children cannot have outpatient treatment? If so, I would have to disagree.

Thank you for raising this concern. We did not mean that SN children cannot have outpatient treatment. The statement on page 2, line 57 has been clarified.

  • Page 2, line 70- a “pre-hospitalization service” is not a standard term internationally. Is this a medical service like a pediatrician? I would agree that these children MUST have preoperative assessment, but I don’t agree that it MUST be with a pediatrician.

The term "pre-hospitalization service" has been replaced with the more correct term "preoperative assessment service" - following your suggestion. We agree with you that this service does not necessarily have to be provided by a pediatrician. Please see lines 84-85, page 2.

  • Section 2.1- it is unusual to list the three groups of pathologies but then start the description of the pathologies in a different order than previously listed (ie. Down syndrome is in group 3 but is the first disease discussed). The order of specific diagnoses should be the same as the group summary.

This is an excellent point. Now, the order of specific diagnoses is the same as the group summary.

  • For Down Syndrome specifically, would summarize the recommendations regarding cervical radiography prior to anesthesia given the propensity towards atlanto-occipital instability.

A summary of cervical spine instability in children with Down syndrome has been added. Please see 2.1.1. section.

  • Tables 2 and 3- the abbreviations “PL” and “GC” are used but haven’t yet been introduced (might just be this particular layout as I know see these abbreviations in section 3, though the tables were on the previous page)
  • Table uses “PL” and the abbreviation in the text is “LP”
  • The tables have variable formatting- some all caps, some lower case (table 5, 2ndrow just says “ger”).

Tables have been reformatted, and the abbreviations in the tables have been extended.

  • The anti-epileptics drugs are known to alter the metabolism of the muscle relaxants as well and this should be mentioned.

Thank you for this suggestion. The 3.1 section has been expanded to include a summary of this topic.

  • Section 3.3- it is definitely possible to do outpatient procedures for patients with SN.

We agree with you. The meaning of our statement was not referring to outpatient procedures for patients with special needs. We aimed to underline the importance to facilitate treatment and assistance by the medical and the nursing staff dedicated to them.

We apologize if we have caused any confusion.

  • The ketamine dose listed here would induce general anesthesia which isn’t really the goal of a premedication.

Barbic et al (Barbic D, Andolfatto G, Grunau B, Scheuermeyer FX, Macewan B, Qian H, Wong H, Barbic SP, Honer WG. Rapid Agitation Control With Ketamine in the emergency Departement:Arandomized Control Trial Ann Emerg Med. 2021 Dec;78(6):788-795. doi: 10.1016/j.annemergmed.2021.05.023) demonstrated that 5 mg/kg intramuscular injection of ketamine provided a significantly short time to adequate sedation, without inducing general anesthesia effects. We added this reference in table 8. We considered the data indicated by Abdel-Ghaffar et al (Abdel-Ghaffar HS, Abdel-Wahab AH, Roushdy MM, Osman AMM. Preemptive nebulized ketamine for pain control after tonsillectomy in children: a randomized controlled trial. Braz J Anesthesiol. 2019 Jul-Aug;69(4):350-357. doi: 10.1016/j.bjan.2019.03.007) and by Mihaljevic et al (Mihaljević S, Pavlović M, Reiner K, Ćaćić M. Therapeutic Mechanism of Ketamine. Psychiatr Danub. 2020 Autumn-Winter;32(3-4):325-333. doi: 10.24869/psyd.2020.325) about nebulized and intravenously administration of ketamine. The references have been added to the manuscript and in the table.

  • I’m unfamiliar with the use of melatonin as a premedication. If the authors can support this with literature, it would enhance the article.

Thank you for raising this concern. A section about Melatonin has been added. Please, see the end of the 3.4.1.2 section.

  • Authors may consider including triazolam (if available in author’s home country) as this is a widely used dental office premedication (where many children with SN are treated).

A paragraph about triazolam has been added. Please, see the 3.4.1.2. section.

  • Page 9, line 323 before is spelled “begore”

We have corrected the spelling error on page 9, line 323 of our documentation.

  • Table 9 has some asterisks by some of the categories but it is unclear why

The asterisks have been deleted.

  • Last line of conclusion- would change “DS” to “SN”

We have replaced it, following your suggestion.

The term “DS” has been replaced with “SN”, as you suggested. We are sorry for this mistake.

Reviewer 2 Report

·        Page 2, line 50- should be “as least traumatic as possible” rather than “less” (same  page 2, line 64)

·        page 2, line 57- is this implying that SN children cannot have outpatient treatment? If so, I would have to disagree.

·        Page 2, line 70- a “pre-hospitalization service” is not a standard term internationally. Is this a medical service like a pediatrician? I would agree that these children MUST have preoperative assessment, but I don’t agree that it MUST be with a pediatrician

·        Section 2.1- it is unusual to list the three groups of pathologies but then start the description of the pathologies in a different order than previously listed (ie. Down syndrome is in group 3 but is the first disease discussed). The order of specific diagnoses should be the same as the group summary.

·        For Down Syndrome specifically, would summarize the recommendations regarding cervical radiography prior to anesthesia given the propensity towards atlanto-occipital instability

·        Tables 2 and 3- the abbreviations “PL” and “GC” are used but haven’t yet been introduced (might just be this particular layout as I know see these abbreviations in section 3, though the tables were on the previous page)

·        Table uses “PL” and the abbreviation in the text is “LP”

·        The tables have variable formatting- some all caps, some lower case (table 5, 2nd row just says “ger”)

·        The anti-epileptics drugs are known to alter the metabolism of the muscle relaxants as well and this should be mentioned

·        Section 3.3- it is definitely possible to do outpatient procedures for patients with SN

·        The ketamine dose listed here would induce general anesthesia which isn’t really the goal of a premedication.

·        I’m unfamiliar with the use of melatonin as a premedication. If the authors can support this with literature, it would enhance the article.

·        Authors may consider including triazolam (if available in author’s home country) as this is a widely used dental office premedication (where many children with SN are treated)

·        Page 9, line 323 before is spelled “begore”

·        Table 9 has some asterisks by some of the categories but it is unclear why

·        Last line of conclusion- would change “DS” to “SN”

Author Response

Comments and Suggestions for Author

Reviewer 2

Thank you for appreciating our work. We are grateful for your constructive criticism and suggestions. We are confident that your suggestions have improved our manuscript.

Page 2, line 50- should be “as least traumatic as possible” rather than “less” (same page 2, line 64)

The term “less” has been replaced with “least”, as you suggested. Please see page 2, lines 53 and 72.

  • page 2, line 57- is this implying that SN children cannot have outpatient treatment? If so, I would have to disagree.

Thank you for raising this concern. We did not mean that SN children cannot have outpatient treatment. The statement on page 2, line 57 has been clarified.

  • Page 2, line 70- a “pre-hospitalization service” is not a standard term internationally. Is this a medical service like a pediatrician? I would agree that these children MUST have preoperative assessment, but I don’t agree that it MUST be with a pediatrician.

The term "pre-hospitalization service" has been replaced with the more correct term "preoperative assessment service" - following your suggestion. We agree with you that this service does not necessarily have to be provided by a pediatrician. Please see lines 84-85, page 2.

  • Section 2.1- it is unusual to list the three groups of pathologies but then start the description of the pathologies in a different order than previously listed (ie. Down syndrome is in group 3 but is the first disease discussed). The order of specific diagnoses should be the same as the group summary.

This is an excellent point. Now, the order of specific diagnoses is the same as the group summary.

  • For Down Syndrome specifically, would summarize the recommendations regarding cervical radiography prior to anesthesia given the propensity towards atlanto-occipital instability.

A summary of cervical spine instability in children with Down syndrome has been added. Please see 2.1.1. section.

  • Tables 2 and 3- the abbreviations “PL” and “GC” are used but haven’t yet been introduced (might just be this particular layout as I know see these abbreviations in section 3, though the tables were on the previous page)
  • Table uses “PL” and the abbreviation in the text is “LP”
  • The tables have variable formatting- some all caps, some lower case (table 5, 2ndrow just says “ger”).

Tables have been reformatted, and the abbreviations in the tables have been extended.

  • The anti-epileptics drugs are known to alter the metabolism of the muscle relaxants as well and this should be mentioned.

Thank you for this suggestion. The 3.1 section has been expanded to include a summary of this topic.

  • Section 3.3- it is definitely possible to do outpatient procedures for patients with SN.

We agree with you. The meaning of our statement was not referring to outpatient procedures for patients with special needs. We aimed to underline the importance to facilitate treatment and assistance by the medical and the nursing staff dedicated to them.

We apologize if we have caused any confusion.

  • The ketamine dose listed here would induce general anesthesia which isn’t really the goal of a premedication.

Barbic et al (Barbic D, Andolfatto G, Grunau B, Scheuermeyer FX, Macewan B, Qian H, Wong H, Barbic SP, Honer WG. Rapid Agitation Control With Ketamine in the emergency Departement:Arandomized Control Trial Ann Emerg Med. 2021 Dec;78(6):788-795. doi: 10.1016/j.annemergmed.2021.05.023) demonstrated that 5 mg/kg intramuscular injection of ketamine provided a significantly short time to adequate sedation, without inducing general anesthesia effects. We added this reference in table 8. We considered the data indicated by Abdel-Ghaffar et al (Abdel-Ghaffar HS, Abdel-Wahab AH, Roushdy MM, Osman AMM. Preemptive nebulized ketamine for pain control after tonsillectomy in children: a randomized controlled trial. Braz J Anesthesiol. 2019 Jul-Aug;69(4):350-357. doi: 10.1016/j.bjan.2019.03.007) and by Mihaljevic et al (Mihaljević S, Pavlović M, Reiner K, Ćaćić M. Therapeutic Mechanism of Ketamine. Psychiatr Danub. 2020 Autumn-Winter;32(3-4):325-333. doi: 10.24869/psyd.2020.325) about nebulized and intravenously administration of ketamine. The references have been added to the manuscript and in the table.

  • I’m unfamiliar with the use of melatonin as a premedication. If the authors can support this with literature, it would enhance the article.

Thank you for raising this concern. A section about Melatonin has been added. Please, see the end of the 3.4.1.2 section.

  • Authors may consider including triazolam (if available in author’s home country) as this is a widely used dental office premedication (where many children with SN are treated).

A paragraph about triazolam has been added. Please, see the 3.4.1.2. section.

  • Page 9, line 323 before is spelled “begore”

We have corrected the spelling error on page 9, line 323 of our documentation.

  • Table 9 has some asterisks by some of the categories but it is unclear why

The asterisks have been deleted.

  • Last line of conclusion- would change “DS” to “SN”

We have replaced it, following your suggestion.

The term “DS” has been replaced with “SN”, as you suggested. We are sorry for this mistake.

Round 2

Reviewer 1 Report

While previous concerns were addressed, many of the same issues remain - chart formatting, word choice, etc.  I did not see the paragraph of non-pharmacological methods actually added to the paper as the authors suggested they did

Author Response

We are sorry if some issues still remain. Tables have been reformatted and the paragraph of non-pharmacological methods have been highlighted.